# Tumor-Infiltrating Lymphocytes as Biomarkers of Treatment Response and Long-Term Survival in Patients with Rectal Cancer: A Systematic Review and Meta-Analysis

**DOI:** 10.3390/cancers14030636

**Published:** 2022-01-27

**Authors:** Adile Orhan, Faisal Khesrawi, Michael Tvilling Madsen, Rasmus Peuliche Vogelsang, Niclas Dohrn, Anne-Marie Kanstrup Fiehn, Ismail Gögenur

**Affiliations:** 1Center for Surgical Science, Zealand University Hospital, Lykkebaekvej 1, DK-4600 Køge, Denmark; faisalkhesrawi@gmail.com (F.K.); mitm@regionsjaelland.dk (M.T.M.); rvo@regionsjaelland.dk (R.P.V.); Niclas.Dohrn@regionh.dk (N.D.); ankf@regionsjaelland.dk (A.-M.K.F.); igo@regionsjaelland.dk (I.G.); 2Department of Surgery, Copenhagen University Hospital, Herlev & Gentofte, Borgmester Ib Juuls Vej 1, DK-2730 Herlev, Denmark; 3Department of Pathology, Zealand University Hospital, Sygehusvej 10, DK-4000 Roskilde, Denmark; 4Institute for Clinical Medicine, University of Copenhagen, Blegdamsvej 3b, DK-2200 Copenhagen, Denmark

**Keywords:** rectal neoplasms, tumor-infiltrating lymphocytes, cancer biomarkers, CD8^+^ T cells, complete response

## Abstract

**Simple Summary:**

This study investigated tumor-infiltrating lymphocytes (TILs) in pretherapeutic biopsies as biomarkers of treatment response and long-term prognosis in patients with locally advanced rectal cancer undergoing neoadjuvant chemoradiotherapy. A systematic review and meta-analysis was performed in accordance with the PRISMA guidelines. The results indicate that it is possible to identify a sub-group of patients with improved treatment response and long-term prognosis by assessing the density of CD8^+^ TILs at the time of diagnosis.

**Abstract:**

Neoadjuvant chemoradiotherapy (NCRT) is indicated in locally advanced rectal cancer (LARC) to downstage tumors before surgery. Watchful waiting may be a treatment option to avoid surgery in patients, obtaining a complete clinical response. However, biomarkers predictive of treatment response and long-term prognosis are lacking. Here we investigated tumor-infiltrating lymphocytes (TILs) in pretherapeutic biopsies as predictive and prognostic biomarkers. A systematic review and meta-analysis was performed in accordance with the PRISMA guidelines. In total, 429 articles were identified, of which 19 studies were included in the systematic review and 14 studies in the meta-analysis. Patients with high pretherapeutic CD8^+^ TILs density had an increased likelihood of achieving a pathological complete response (RR = 2.71; 95% CI: 1.58–4.66) or a complete or near-complete pathological treatment response (RR = 1.86; 95% CI: 1.50–2.29). Furthermore, high CD8^+^ TILs density was a favorable prognostic factor for disease-free survival (HR = 0.57; 95% CI: 0.38–0.86) and overall survival (HR = 0.43; 95% CI: 0.27–0.69). CD3^+^, CD4^+^, and FOXP3^+^ TILs were not identified as predictive or prognostic biomarkers. Thus, assessing pretherapeutic CD8^+^ TILs density may assist in identifying patients with increased sensitivity to NCRT and favorable long-term prognosis.

## 1. Introduction

Neoadjuvant chemoradiotherapy (NCRT) is indicated in patients with locally advanced rectal cancer (LARC) to downstage tumors before surgery. The current standard of care consists of short or long-course radiotherapy combined with single-agent fluoropyrimidine-based chemotherapy or in combination with oxaliplatin. Following NCRT, a complete clinical response is observed in 10–40% of patients [1]. Inadequate treatment response is associated with an unfavorable long-term prognosis [2]. For patients obtaining a complete clinical response, watchful waiting may be an alternative treatment option allowing certain patients to undergo an organ-preserving strategy and avoid major surgery [3]. The achievement of complete response is affected by the initial clinical UICC stage, pathological subtype, pretherapeutic carcinoembryonic antigen levels, and other currently unidentified factors [1,4]. Some studies have identified the density of tumor-infiltrating lymphocytes (TILs) as a predictive marker of treatment response and long-term prognosis in patients with LARC undergoing NCRT [5,6]. Assessment of TILs is not yet included as a part of the routine histopathological examination. However, in research, immunohistochemical (IHC) analysis has been used to assess TILs density and specific subgroups of lymphocytes. CD3^+^, CD4^+^, CD8^+^, and FOXP3^+^ TILs are among the most commonly evaluated markers [7]. High levels of CD3^+^, CD4^+,^ and CD8^+^ TILs in the tumor microenvironment are reported to be a signature of active antitumor immunity. In contrast, high levels of FOXP3^+^ TILs have been associated with an immunosuppressive tumor microenvironment [8,9,10,11]. Infiltration of TILs, especially CD8^+^ T cells, in pretherapeutic tumor tissue has been correlated with improved survival in different gastrointestinal cancers, including gastric cancer, hepatocellular carcinoma, and pancreatic cancer [12,13,14].

This review and meta-analysis examines TILs in pretherapeutic biopsies and their significance in the prediction of treatment response and long-term prognosis in patients with LARC undergoing NCRT and surgical resection.

## 2. Materials and Methods

### 2.1. Protocol and Registration

This study was designed according to the PRISMA-P guidelines [15] for systematic review and meta-analysis. A PRISMA checklist is available in Appendix A. The protocol was registered at the Center for Open Science under the digital object identifier: osf.io/npgux (accessed on 18 August 2020)

### 2.2. Search Strategy

The literature search was conducted using PubMed, Embase, Cochrane Library, and Web of Science. A Population-Intervention-Comparator-Outcome (PICO)-based search strategy was applied. The population of interest was restricted to Union for International Cancer Control (UICC) stage II–IV human rectal cancers undergoing NCRT before curative intended surgical resection. Assessment of TILs density by at least one of the following IHC markers CD3, CD4, CD8, or FOXP3 in pretherapeutic biopsies was the intervention. As comparators, we considered patients with low versus high TILs density and patients with treatment response versus patients with no response. The outcomes of interest were pathological complete response (pCR) and pathological treatment response (pTR), overall survival (OS), and disease-free survival (DFS) or recurrence-free survival (RFS). pCR and pTR were evaluated histopathologically in the surgical specimen using the tumor regression grade (TRG). There was no restriction regarding the TRG system applied for evaluation.

The search was restricted to human trials and English language articles. Studies with a sample size below 30 were deemed too small to provide relevant information and were excluded. Case studies, reviews, conference abstracts, commentaries, and letters to editors were also excluded. No search restrictions regarding publication date or study design were included in the final search string. The search strategy is summarized with reference to the PICO process in Appendix A. The last search was performed on 12 December 2020.

### 2.3. Data Management and Extraction

Search results from the four databases were imported into EndnoteX9 desktop application (Clarivate, Philadelphia, PA, USA) for duplicate removal. After removing duplicates, the articles were imported into Covidence web application (Veritas Health Innovation, Melbourne, Australia) for article screening. Subsequently, two reviewers, FK and AO, independently screened articles at the title and abstract level, followed by full-text screening. The relevance of items was decided based on the predefined eligibility criteria. Any disagreement was resolved by a consensus decision with the inclusion of a third reviewer (RV).

Study data including UICC tumor stage, treatment type, evaluated subgroups of lymphocytes, TILs density, and quantification method (manual vs. digital), as well as the cut-off value for low vs. high densities (median, mean, combined scores, or others), TRG and TRG system were extracted and registered. Corresponding authors of studies with insufficient or missing data were contacted by email, and in case of no reply, studies were excluded from the meta-analysis.

### 2.4. Outcomes and Subgroups

The primary outcome was pCR, defined as no residual cancer cells left in the surgical specimen according to the Dworak, Mandard, and AJCC TRG systems [16,17,18]. Studies using other TRG systems to assess pCR were included in the meta-analyses if a pCR did not vary from this definition. Patients not achieving a pCR were grouped as non-responders and used as comparators in the dichotomous analyses.

The secondary outcome was pTR, defined as a complete or near-complete tumor regression corresponding to a Dworak TRG 3-4, Mandard TRG 1-2, or AJCC TRG 0-1. Studies using other TRG systems were included in the meta-analyses of pTR if the definition did not vary significantly. This outcome was selected as many studies had not differentiated between these two subgroups with no or only minute foci of residual tumor. Patients not achieving pTR were grouped as non-responders and used as comparators in the dichotomous analyses. 

OS and DFS/RFS were also included as secondary outcomes reflecting long-term prognosis. The definitions of OS and DFS were based on The National Cancer Institute’s Dictionary of Cancer Terms. OS was defined as the time between diagnosis of rectal cancer and the date of death regardless of cause. DFS was defined as the time elapsing from the date of surgery to the first symptoms of recurrent disease or death. We defined RFS as equivalent to DFS, and studies with data on RFS were included in the meta-analysis for DFS. Patients with stage IV disease, and treated with non-curative intent, were excluded from the meta-analyses of survival outcomes. 

### 2.5. Data Extraction

Hazard ratios (HR) with corresponding 95% confidence intervals (CIs) of high vs. low density of TILs were extracted. In studies where only the Kaplan-Meier (KM) plots were available and HR was not presented, the HR was estimated through the KM plots using the DigitizeIt desktop application and methods as described by Tierney et al. [19].

### 2.6. Statistical Analysis

The meta-analyses were conducted separately for each subgroup of TILs and outcomes of interest. All meta-analyses were performed in R statistical software (version 3.4.3, R Foundation for Statistical Computing) applying the Meta [20], Metagen [21], and Metafor [22] packages. Dichotomous outcomes were analyzed using a random-effects model calculating risk ratios (RR) using the Mantel-Haenszel method. Sidik-Jonkman was applied for tau^2^ estimation and Hartung-Knapp adjustment of 95% CI for the random-effect model [23]. When applicable, prediction intervals supplying an estimate of the expected effect sizes on future studies were calculated [24]. Heterogeneity was assessed by Chi^2^ testing and I^2^ statistics. A random-effect model using the inverse variance to estimate HR was applied in meta-analyses of time to event outcomes. The estimate of variance in the meta-analysis was based on upper and lower 95% CI as described by Altman et al. [25], and analysis was performed as described for dichotomous outcomes.

### 2.7. Risk of Bias of Individual Studies

The risk of bias within studies was assessed using the Newcastle-Ottawa Scale (NOS) for cohort studies [26]. The NOS examines three domains of a study: patient selection, cohort comparability, and outcome assessment. The maximum attainable NOS score was nine points.

### 2.8. Level of Evidence

The level of evidence was evaluated using a modified GRADE approach for observational studies. The modified version of the GRADE approach can specifically be used in systematic reviews of prognostic markers. According to this approach, the cumulative evidence can be rated as either of high, moderate, low, or very low quality [27]. Each meta-analysis result starts as high-quality evidence. Several limitations can reduce the quality of evidence (risk of bias, inconsistency of results, indirectness, imprecision, and publication bias). Limitations considered not serious do not downgrade evidence. Serious limitations downgrade quality with one level, and very serious limitations downgrade quality with two levels. In contrast, strengths of evidence (large effect size and dose-response gradient) can upgrade the level of evidence. Each strength of evidence upgrades quality with one level.

## 3. Results

### 3.1. Search Results

The literature search identified 429 articles (Figure 1). After removing duplicated articles, a total of 307 studies were screened for title and abstract, ultimately revealing 86 studies for further full-text screening. After the full-text screening was completed, 19 studies were eligible for inclusion in the systematic review [5,6,9,10,11,28,29,30,31,32,33,34,35,36,37,38,39,40,41]. Of these, 14 studies were eligible [5,6,9,10,28,29,30,31,32,33,34,35,36,37] and five studies were ineglible for meta-analysis [11,38,39,40,41] (Figure 1). The five studies ineligible in the meta-analysis were due to the following reasons: Three studies were excluded from the meta-analysis as the subgroups of TILs were not dichotomized into high or low-density groups for each individual marker [38,39,41], whereas two were ineligible as the studies included patients that did not undergo curatively intended treatment [11,40].

### 3.2. Study Characteristics

The included studies were conducted in Asia (*n* = 12), Europe (*n* = 4), Middle East (*n* = 2), and Australia (*n* = 1). The studies were published between 2011 and 2020. All studies were retrospective cohort studies. The combined cohort, based on all the eligible studies in the systematic review, consisted of 2034 patients. Most studies included UICC stage II–III cancers, although three studies also included stage IV cancers eligible for curative intended therapy [6,11,40]. All patients underwent NCRT before curative intended surgery. Long-course NCRT was used in 16 studies (84%), whereas two studies did not specify the type of NCRT [5,38], and one study used short-course NCRT [28]. All studies assessed TILs in the diagnostic biopsies using IHC stainings. Different cut-off values were used for dichotomizing the TILs into groups of high vs. low densities. Most studies used the median TILs value, yet mean values, combined scores, and unspecified values were also used. Fifteen studies used Dworak, Mandard, or AJCC for the assessment of TRG. Two studies used the Japanese Colorectal Cancer Classification (JCCC) TRG system [6,30], while two studies did not specify which TRG system they used [31,38] (Table 1).

### 3.3. Pathological Complete Response (pCR)

Six studies were eligible for the meta-analysis of the primary outcome, pCR. In four studies, Dworak, Mandard, or AJCC [29,33,35,36] were used, while two studies used other TRG systems [6,31]. The eligible studies provided information regarding high versus low density of CD8^+^ TILs on 692 patients in total. Of the patients included in the meta-analysis, 282 patients (41%) were categorized as having a high CD8^+^ TILs density and 410 patients (59%) with having a low CD8^+^ TILs density. The categorization of patients in either the high or low CD8^+^ T cell expression groups was based on different cut-off estimation methods in the six studies included in the meta-analysis. Three studies did not specify the cut-off estimation [6,29,36], one of the studies based the cut-off on the precalculated median TIL value in the cohort [33], whereas one used 11 cells per high power field as the cut-off value [35], and the last study used a receiver operating characteristics (ROC) curve [31] (Table 1). Patients with high CD8^+^ TILs density had an increased likelihood of achieving a pCR compared to patients with a low CD8^+^ TILs density (RR = 2.71; 95% CI: 1.58–4.66) (Figure 2).

Meta-analysis on the association between pretherapeutic CD3^+^, CD4^+^, and FOXP3^+^ TILs and pCR could not be performed due to a lack of studies reporting data on these variables. 

### 3.4. Pathological Treatment Response (pTR)

Overall, 10 studies were eligible for the meta-analyses of the secondary outcome, pTR. Nine studies assessed TRG according to Dworak, Mandard, or AJCC [5,9,10,28,29,32,33,35,37]. Matsutani et al. used JCCC and included TRG 1b-3 in the group of pTR [30].

#### 3.4.1. CD8^+^ TILs

Nine studies providing data on 1073 patients in total were included in the meta-analysis on pretherapeutic CD8^+^ TILs density and pTR [9,10,28,29,30,32,33,35,37]. In total, 491 patients (46%) were categorized with high CD8^+^ TILs levels, while 582 (54%) of the patients were categorized with low CD8^+^ TILs levels. Again, the differentiation between high versus low levels of the examined lymphocyte subsets was primarily based on a precalculated median value of the examined cohort in each of the eligible studies. Patients with high levels of CD8^+^ TILs had an increased likelihood of obtaining a pTR (RR = 1.86; 95% CI: 1.50–2.29) (Figure 3).

#### 3.4.2. CD3^+^ TILs

Two studies with 191 patients in total investigated the association between pretherapeutic CD3^+^ TILs density and pTR [5,10]. Of the total, 102 (53%) patients were categorized as CD3^+^ TILs high, whereas the remaining 89 (47%) patients had low CD3^+^ lymphocyte infiltration. High density of CD3^+^ TILs was not significantly associated with any changes in pTR (RR = 1.63; 95% CI: 0.35–7.69, *p* = 0.16) (Appendix A).

#### 3.4.3. CD4^+^ TILs

Four studies consisting of 342 patients altogether, evaluated the association between pretherapeutic CD4^+^ TILs density and pTR [9,30,32,37]. High CD4^+^ TILs were reported in 175 patients (51%), while the remaining 167 (49%) of patients in the total cohort had low expression of CD4^+^ TILs. CD4^+^ TILs density was not correlated with significant changes in pTR (RR = 1.23; 95% CI: 0.83–1.82, *p* = 0.19) (Appendix A).

#### 3.4.4. FOXP3^+^ TILs

Four studies with 294 patients in total examined the association between pretherapeutic FOXP3^+^ TILs density and pTR [9,28,30,32]. Overall, 129 patients (44%) were categorized as having high FOXP3^+^ TILs expression, while the remaining 165 (56%) had low expression. The meta-analysis revealed no significant association between FOXP3^+^ TILs density and pTR (RR = 0.85; 95% CI: 0.20–3.58, *p* = 0.74) (Appendix A).

### 3.5. Overall Survival (OS)

Four studies providing data on 360 patients altogether examined the association between CD8^+^ TILs and OS [9,10,31,34]. Of the total number of patients, 170 (47%) patients had high CD8^+^ lymphocyte infiltration and 190 (53%) patients had low CD8^+^ lymphocyte infiltration. Meta-analysis of time-to-event data revealed that a high CD8^+^ TILs density was a significantly favorable prognostic factor for OS (HR = 0.43; 95% CI: 0.27–0.69) (Figure 4).

### 3.6. Disease-Free Survival (DFS)

#### 3.6.1. CD8^+^ TILs

Six studies consisting of 716 patients overall evaluated the association between CD8^+^ TILs levels and DFS [9,10,28,31,33,34]. In sum, 349 (49%) and 367 (51%) of the patients were categorized with high and low CD8^+^ TILs density, respectively. Time-to-event analysis revealed that a high CD8^+^ TILs density was significantly associated with improved DFS (HR = 0.57; 95% CI: 0.38–0.86) (Figure 4).

#### 3.6.2. FOXP3^+^ TILs

Three studies with a sum of 193 patients examined whether the pretherapeutic FOXP3^+^ TILs level was a prognostic factor for DFS [9,28,31]. Collectively, 98 (51%) of the patients were categorized as having a high FOXP3^+^ TILs level, while 95 (49%) of the patients were categorized as having a low FOXP3^+^ TILs level. The FOXP3^+^ TILs density was not associated with any significant changes in DFS (HR = 1.66; 95% CI: 0.17–16.32) (Appendix A).

### 3.7. Risk of Bias of Individual Studies

Using the NOS scoring system, four of the included studies in the systematic review achieved seven points out of a maximum of nine points [5,31,35,40]. Ten studies achieved eight points [6,9,10,11,29,30,32,33,36,38]. Five studies achieved the maximum NOS score of nine points [28,34,37,39,41]. 

Ten studies were downgraded in the comparability section as the studies did not control for confounders [5,9,10,11,30,31,32,35,38,40]. Seven studies were downgraded in the outcome section as the studies did not report whether TILs were assessed by independent and blinded assessors [5,29,31,33,35,36,40]. A detailed description is available in Appendix A.

### 3.8. Level of Evidence

Using the GRADE approach, the level of evidence was rated as moderate on all outcomes for CD8^+^ TILs. For pCR and OS, the quality of the cumulative evidence was downgraded two levels due to potential publication bias and imprecision but upgraded one level due to a large effect size. For pTR and DFS, the quality of evidence was downgraded one level due to potential publication bias. The quality of the cumulative evidence for the results related to the CD3^+^, CD4^+,^ and FOXP3^+^ TILs subsets was rated as low or very low due to multiple limitations. Publication bias could not be statistically assessed by funnel plots due to the limited number of studies, and therefore, all results were downgraded one level. A detailed overview is available in Appendix A.

Table 2 provides an overview of the abovementioned results, including the quality of the cumulative evidence based on the modified GRADE approach.

## 4. Discussion

In this systematic review and meta-analysis, we investigated the pretherapeutic TILs density as a predictive marker of treatment response and long-term prognosis in patients with LARC undergoing NCRT and surgery. The meta-analysis found that patients with a high pretherapeutic CD8^+^ TILs density had an increased likelihood of achieving pCR and pTR. Correspondingly, patients with a high pretherapeutic CD8^+^ TILs density had a favorable DFS and OS compared to patients with low CD8^+^ TILs infiltration. Our findings are consistent with previously published meta-analyses that have correlated high CD8^+^ TILs density to a favorable prognosis for patients with colorectal cancer [7,8].

It is hypothesized that patients with rectal cancer having a high pretherapeutic TILs density are more likely to have a good response to NCRT, as the preexisting antitumor immunity is enhanced by NCRT. This hypothesis is supported by preclinical studies showing that CRT alters the tumor microenvironment (TME) in favor of antitumor immunity [42,43]. Tumor cell death following CRT causes a release of tumor neoantigens and pro-inflammatory mediators into the TME, increasing T-cell infiltration, T-cell elimination of cancer cells, and facilitating further adaptive antitumor responses [44,45,46,47].

Combining NCRT with immunotherapy may additionally enhance the T-cell mediated antitumor response [48,49,50]. Multiple clinical trials are currently investigating whether the addition of immunotherapy to NCRT regimens increases pCR rates in patients with LARC (ClinicalTrials.gov (accessed on 6 January 2022): NCT04017455, NCT03127007, NCT03102047, NCT03854799, and NCT03921684). Recently, a phase II randomized clinical trial examined the efficacy and safety of adding the immune check-point inhibitor (ICI), pembrolizumab, to standard NCRT for the treatment of rectal cancer [51]. The study found pCR in 31.9% and 29.4% of the patients in the pembrolizumab arm and control arm, respectively. The findings did not reach statistical significance, and the percentage of grade 3 and 4 adverse events were slightly higher in the pembrolizumab arm [51]. However, the study did not examine the pretherapeutic TILs level, the amount of tumor stroma, the number of cancer-associated fibroblasts (CAFs), or other TME factors associated with a tumor phenotype that negatively impacts response rates and survival measures [52,53]. The composition of the TME highly influences the infiltration, survival, and proliferation of TILs. Fibrotic tissue surrounding cancer cells may prevent direct cell-to-cell contact between lymphocytes and cancer cells, thus hindering the killing mechanism of malignant cells by lymphocytes. Also, hypoxia in the TME may lead to apoptosis of TILs, thereby reducing their number and effectivity in the tumor tissue [53].

Examining the density of TILs as well as the composition of the TME prior to treatment with ICI have the potential to further assist the clinicians in the selection of patients that may benefit from this treatment. However, tumor sampling by using single biopsies has been associated with limitations [54]. The tumoral immune infiltrate and the composition of the TME may be better evaluated with multiple biopsies or by selecting a larger tumor area for sampling [55]. These considerations should also be taken into account when analyzing and interpreting the TILs density and the overall composition of the TME. In continuation, the Immunoscore has been internationally validated as a prognostic tool for colon cancer [56,57]. The Immunoscore is a scoring system categorizing patients as low, intermediate, or high based on the density of CD8^+^ and CD3^+^ T cells and their abundance in the center and margin of the tumor.

The international validation studies on the Immunoscore in colon cancer have found the method to be a reliable estimate of recurrence in patients with colon cancer with superiority to several routinely assessed parameters as histopathological differentiation, microsatellite instability (MSI) status, and vascular and perineural invasion [56,57]. Likewise, a diagnostic biopsy-adapted immunoscore has been found to be a predictive method for both survival and neoadjuvant treatment response in patients with LARC undergoing surgery [41]. This approach may also assist in selecting patients with LARC that may benefit from a “watchful waiting” regimen when planning the treatment in the oncological and surgical setting.

Limitations of this study should also be addressed. Firstly, the GRADE assessment revealed low to moderate levels of evidence. For many of the examined outcomes and TILs subgroups examined in this study, the number of data eligible for the meta-analysis was limited and the results for some of the outcomes and TILs subgroups were thus based on small sample sizes. The limited number of studies also prevented proper assessment of the risk of publication bias. Secondly, our evaluation of the risk of bias in individual studies revealed several limitations. All the included studies were retrospective cohort studies creating a risk of selection or information bias. Some of the included studies did not clarify if experienced and blinded pathologists assessed the TRG and TILs density. Evaluation of TRG is associated with significant interobserver variability [58], and only a few of the included studies have addressed this problem. Our meta-analyses combined TRG assessment using various TRG systems. Although the TRG systems are quite similar, minor differences exist with the risk of increasing the imprecision of the results. Furthermore, most studies assessed TILs density manually and not digitally. The manual assessment of TILs is associated with interobserver variability, which may result in a considerate inconsistence in the reported data. Complete pathological response is usually not associated with interobserver variability. However, most pathologists find tumor regression grading difficult. Furthermore, the histological examination including number of examined sections from each tumor block is not uniform among pathologists [59]. In the United Kingdom, it is required to examine additional deeper sections from each block before reporting a diagnosis of pCR [60], while this is not mentioned as obligatory in the recommendations by the International Collaboration of Cancer Reporting (ICCR) [61].

Likewise, different cut-off values were used to classify the patients into either high or low TILs density groups in the included studies, creating an inconsistence in the cumulative evidence. Most of the included studies used the precalculated median value of the study cohort as the cut-off value to classify patients into either high or low lymphocyte infiltration groups. The median value of TILs may vary, not just across different cancer types, but also in different cohorts of patients with the same cancer. This variability in the categorization of patients should be taken into account when interpreting the results of this meta-analysis. Standardizing the cut-off value of TILs for different cancer types may thus assist in creating a more consistent classification of tumors as having high or low TILs levels. Consensus concerning a standardized cut-off value optimized to predict outcomes as well as an exact definition of the area where to perform the counts would probably result in a more homogenous group and studies would be easier to compare, especially when conducting meta-analyses.

Lastly, our meta-analyses were performed on dichotomized data. While dichotomization of values simplifies statistical analysis and clinical application, it hinders the possibility to evaluate a dose-response relationship between the degree of immune infiltration and the degree of tumor regression. Proving a dose-response relationship would, in theory, strengthen the validity of the findings as outlined in the GRADE approach.

Despite the limitations, the findings of this meta-analysis indicate that assessment of pretherapeutic TILs may identify a subgroup of patients with LARC that may benefit from NRCT with improved treatment response and a favorable long-term prognosis as well as patients that may benefit most from a watchful waiting regimen. Thus, assessment of CD8^+^ TILs at time of diagnosis may, among other clinical-biological markers and clinical decision support tools, assist clinicians in selecting patients who may benefit from NCRT and immune-enhancing treatment regimens, establishing a more personalized approach to the treatment of LARC. Further clinical studies examining the association between pre- and post-therapeutic TILs density and improved prognosis, as well as the survival benefit of adding ICIs to standard neoadjuvant therapy for LARC should be conducted.

## 5. Conclusions

High pretherapeutic CD8^+^ TIL levels were associated with pCR and pTR as well as improved DFS and OS in patients with LARC undergoing NCRT, suggesting the density of CD8^+^ TILs in the pretherapeutic biopsies may serve as an important predictive and prognostic biomarker of treatment response and survival.

The same results could not be obtained for CD3^+^ or CD4^+^ TILs. Likewise, the findings related to FOXP3^+^ TILs did not reveal any significant results. However, the results on these TILs subtypes were based on limited data. Further examination of the predictive and prognostic value of pretherapeutic TIL levels in rectal cancer is thus needed. Especially, prospective studies using reproducible digital and consistent quantification methods of both TILs and TRG are highly relevant when conducting future studies.

## Figures and Tables

**Figure 1 cancers-14-00636-f001:**
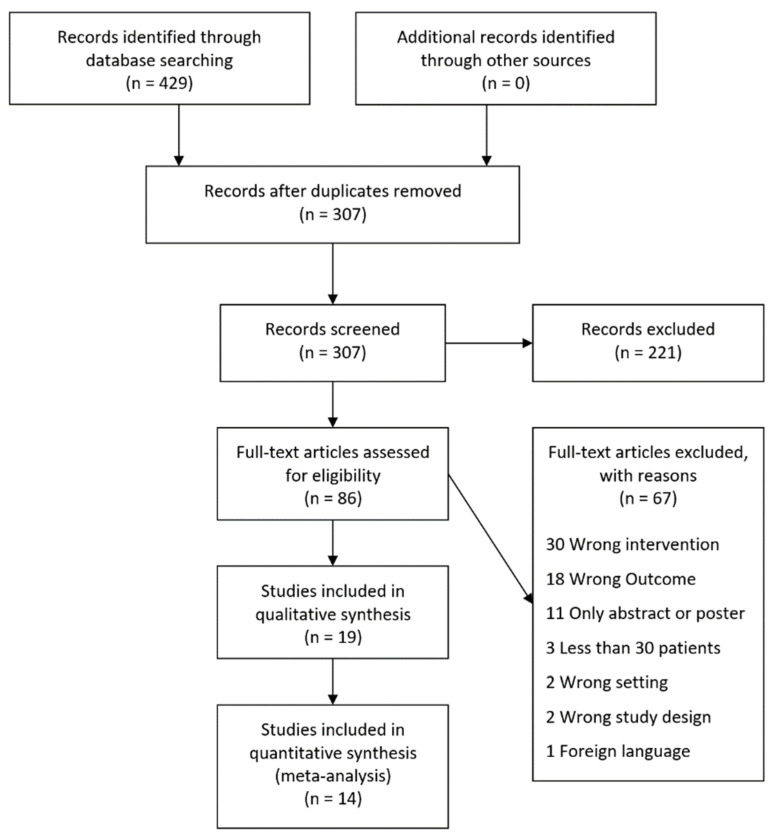
PRISMA flowchart. The literature search was performed using PubMed, Embase, The Cochrane Library, and Web of Science.

**Figure 2 cancers-14-00636-f002:**
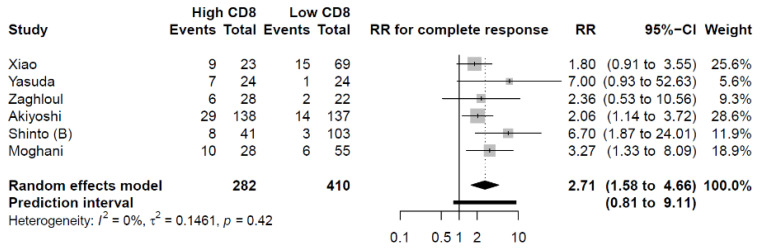
The relationship between CD8^+^ TILs density and pCR. Patients with high pretherapeutic CD8^+^ TILs density had a significantly increased likelihood of achieving a pCR. RR: Risk Ratio.

**Figure 3 cancers-14-00636-f003:**
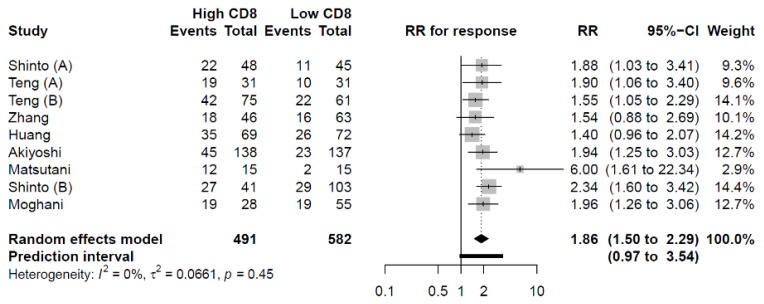
The relationship between CD8^+^ TILs density and pTR. Patients with high CD8^+^ TILs density had a significantly increased likelihood of achieving a pTR. RR: Risk Ratio.

**Figure 4 cancers-14-00636-f004:**
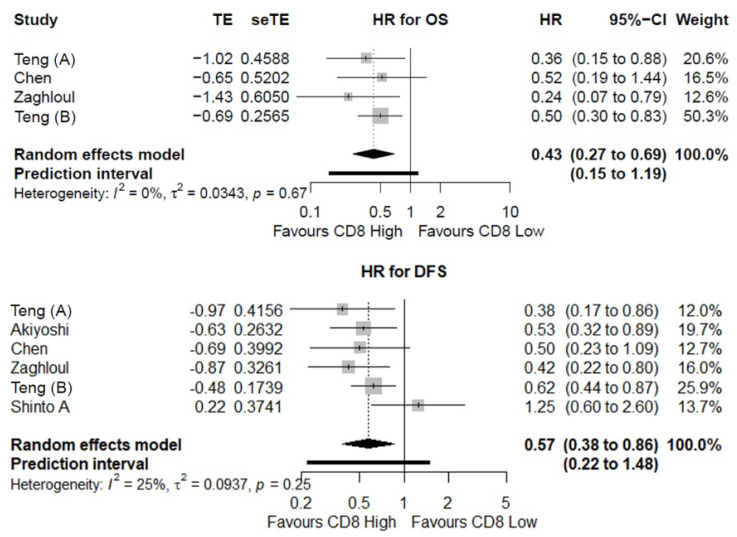
The prognostic value of CD8^+^ TILs. Meta-analyses of time-to-event data indicated a favorable overall survival (OS) and disease-free survival (DFS) for patients with high pretherapeutic CD8^+^ TILs density. HR: Hazard Ratio.

**Table 1 cancers-14-00636-t001:** Main characteristics of the included studies. The density of TILs was assessed in the diagnostic biopsy using IHC with antibodies targeting specific subgroups of TILs. Different cut-off values were used to separate patients into high vs. low density of TILs. Treatment response was evaluated histopathologically using TRG. Treatment response was defined as pCR or pTR in most studies.

Author	UICC Stage	TILs Density Cut-Off	TRG System	Treatment Response Defined as	Inclusion in Meta-Analysis
Anitei et al. [5]	II–III	Median	Dworak	TRG 3-4	CD3 (pTR)
Yasuda et al. [6]	II–IV	Unspecified	JCCC	pCR	CD8 (pCR)
Teng et al. (A) [9]	II–III	Median	Dworak	TRG 3-4	CD4 (pTR) FOXP3 (pTR, DFS) CD8 (pTR, OS, DFS)
Teng et al. (B) [10]	II–III	Median	Dworak	TRG 3-4	CD3 (pTR) CD8 (pTR, OS, DFS)
McCoy et al. [11]	II–IV	Median	Dworak	TRG 3-4	none
Shinto et al. (A) [28]	II–III	Median	Dworak	TRG 3-4	FOXP3 (pTR, DFS) CD8 (pTR, DFS)
Shinto et al. (B) [29]	II–III	Unspecified	Dworak	pCR, TRG 3-4	CD8 (pCR, pTR)
Matsutani et al. [30]	II–III	Median	JCCC	TRG 1b-3	CD4 (pTR) FOXP3 (pTR) CD8 (pTR)
Zaghloul et al. [31]	II–III	ROC	Unspecified	pCR	FOXP3 (DFS) CD8 (pCR, OS, DFS)
Zhang et al. [32]	II–III	Mean	Dworak	TRG 3-4	CD4 (pTR) FOXP3 (pTR) CD8 (pTR)
Akiyoshi et al. [33]	II–III	Median	Dworak	pCR, TRG 3-4	CD8 (pCR, pTR, DFS)
Chen et al. [34]	II–III	Unspecified	Dworak	TRG 3-4	CD8 (OS, DFS)
Moghani et al. [35]	II–III	11 cells/high power field	AJCC	pCR, TRG 0-1	CD8 (pCR, pTR)
Xiao et al. [36]	II–III	Unspecified	Mandard	pCR	CD8 (pCR)
Huang, Y et al. [37]	II–III	Unspecified	AJCC	TRG 0-1	CD4 (pTR) CD8 (pTR)
Mirjolet et al. [38]	II–III	None	Unspecified	Unspecified	none
Huang, A et al. [39]	II–III	CD3/CD8 combined	Dworak	TRG 3-4	none
Rudolf et al. [40]	II–IV	Median	Dworak	Unspecified	none
Sissy et al. [41]	II–III	CD3/CD8 combined	Dworak	Unspecified	none

UICC: Union for International Cancer Control. AJCC: American Joint Committee on Cancer. JCCC: Japanese Colorectal Cancer Classification. TRG: Tumor regression grade. pCR: Pathological complete response. pTR: Pathological treatment response.

**Table 2 cancers-14-00636-t002:** This table provides an overview of the results of the meta-analyses including a point estimate of the risk ratio (in relation to pathological response) and the hazard ratio (in relation to survival) with corresponding CI. The level of evidence has also been included in the table. The meta-analyses evaluated the association between pretherapeutic TILs density and treatment response to NCRT and long-term prognosis.

Outcome	Biomarker	Studies	*n*	Point Estimate	Lower 95% CI	Higher 95% CI	I^2^	GRADE Level of Evidence
pCR	CD8^+^ TILs	6	692	2.71	1.58	4.66	0%	Moderate
pTR	CD3^+^ TILs	2	191	1.63	0.35	7.69	0%	Low
	CD4^+^ TILs	4	342	1.23	0.83	1.82	0%	Low
	CD8^+^ TILs	9	1073	1.86	1.50	2.29	0%	Moderate
	FOXP3^+^ TILs	4	294	0.85	0.20	3.58	76%	Very low
DFS	CD8^+^ TILs	6	716	0.57	0.38	0.86	25%	Moderate
	FOXP3^+^ TILs	3	193	1.66	0.17	16.32	62%	Very low
OS	CD8^+^ TILs	4	360	0.43	0.27	0.69	0%	Moderate

pCR: Pathological complete response. pTR: Pathological treatment response. DFS: Disease-free survival. OS: Overall survival. *n*: number of patients. CI: Confidence interval.

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
