# Peer review of "Tumor-Infiltrating Lymphocytes as Biomarkers of Treatment Response and Long-Term Survival in Patients with Rectal Cancer: A Systematic Review and Meta-Analysis"

_cancers, 2022, doi:10.3390/cancers14030636_

Round 1
Reviewer 1 Report
The study is worthy to share.
however the weakness are as follow:
1/the number of publications included to this analysis was relatively low
2/it is probable that the methods used for lymphoid cells comparison were different
3/ thus the possible use of the results of this study in practice is low
Author Response
We would like to thank the reviewer for both the review of the manuscript and for the comments.
1/the number of publications included to this analysis was relatively low
We agree that the number of publications that could be included in this meta-analysis was unfortunately relatively low. This was however due to only a small number of studies being eligible for inclusion in the meta-analysis. We believe the inclusion criteria was necessary to have comparability between the studies regarding methodologies and outcomes.
2/it is probable that the methods used for lymphoid cells comparison were different
The methods used for lymphocyte infiltration comparison were different in the included studies. Mainly, the median value of TILs in the study cohort was used as the cut-off value to group patients in either high or low TIL density groups. However, some of the studies used the mean or predefined thresholds. We have addressed this issue as a limitation of the study at page 12, line 413-420.
3/ thus the possible use of the results of this study in practice is low
Even though this study had limitations in regard to the total number of eligible articles and differences in the methods for TILs quantification in the articles, the results on pathological response and survival outcomes in relation CD8+ T cell infiltration were significant. Similar results on CD8+ TILs density and survival outcomes have been described in other cancer types such as in pancreatic cancer. The CD8+ T cell infiltration in different cancers is thus of importance and may serve as an important prognostic and predictive marker in the near future. This study, together with other relevant studies on TILs, emphasizes the use of CD8+ TILs quantification in the future for better prognostication and treatment selection. We hope this study will be practice changing when the results are interpreted together with other existing data on CD8+ TILs in rectal cancer and in different cancers.
We have marked all the new changes with red in the resubmitted manuscript.
Once again, we would like to thank both reviewers for the useful comments. We believe the changes suggested from the reviewers have improved the manuscript considerably.
Best regards and on behalf of the author group
Adile Orhan
Reviewer 2 Report
I have read with interest the manuscript entitled “Tumor-infiltrating lymphocytes as biomarkers of treatment response and long-term survival in patients with rectal cancer: a systematic review and meta-analysis”. The work is well performed and the methodology used is correct. The subject is of interest and could be very useful in clinical practice.
I have some comments to make:
The results section should be better explained and structured. From the results it is difficult to determine the cut-offs for TIL’s to be considered high or low, and its range. Although the tables are more complete, each of the results subtitle should be better developed and explained. In some results it is difficult to determine the number of patients assessed, and many other details relevant to the study and that result.
Indeed, each of the results and correlation given comes out from different studies, and the number of cases is very different, as is the relevance of each of the statements. For some of the results, the data were very limited, and it is difficult to determine if they come each from different cohorts or not. The method of assessment of the TIL count in the different studies should be developed further. It is true that regression grades and methods or scales used are not very important in the way that complete regression is the most important outcome and it is easy to assess for a pathologist. Nevertheless, it should be better explained.
Author Response
We would like to express our sincerest thanks to the reviewer for the comments.
The results section should be better explained and structured. From the results it is difficult to determine the cut-offs for TIL’s to be considered high or low, and its range. Although the tables are more complete, each of the results subtitle should be better developed and explained. In some results it is difficult to determine the number of patients assessed, and many other details relevant to the study and that result.
The results section has now been revised. The presentation of the results is more structured and the results in each subtitle are explained better, including presentation of number of patients in total and in each group (high versus low TILs) with percentages in parenthesis. We have also explained the methods of assessment of the TIL count in the different studies in both the results section and addressed this matter in the discussion section. The changes made are highlighted in red and can be found at:
- page 4, line 181
- page 5, line 195
- page 6, line 223-230
- page 7, line 237
- page 7, line 247-253
- page 7, line 260-264
- page 8, line 268-304
- page 9, line 312
Indeed, each of the results and correlation given comes out from different studies, and the number of cases is very different, as is the relevance of each of the statements. For some of the results, the data were very limited, and it is difficult to determine if they come each from different cohorts or not.
We have addressed both the limited number of studies as well as the differences in TILs and TRG quantification methods in the discussion section of the manuscript at page 12, line 397-400, as well as in the conclusion section of the manuscript at page 12, line 444-448.
The method of assessment of the TIL count in the different studies should be developed further. It is true that regression grades and methods or scales used are not very important in the way that complete regression is the most important outcome and it is easy to assess for a pathologist. Nevertheless, it should be better explained.
We have now further explained the cut-off values of TILs used in the included studies at page 6, line 226-228, and have discussed the problems that may be related to a non-standardized cut-off value across the studies. We have also discussed this issue further in the discussion section at page 12, line 413-420, when emphasizing the limitations of the study. At page 12, line 446-448, we have additionally encouraged to further research on the matter in hope of future studies leading to a more standardized quantification method of TILs.
In addition, we have explained the assessment of pathological complete response further at page 12 line 415-421 and have added three new references (marked in red in the reference list).
We have marked all the new changes with red in the resubmitted manuscript.
Once again, we would like to thank both reviewers for the useful comments. We believe the changes suggested from the reviewers have improved the manuscript considerably.
Best regards and on behalf of the author group
Adile Orhan
Round 2
Reviewer 2 Report
The manuscript has been improved and I have no further comments.